# A Review of Electromagnetic Shielding Fabric, Wave-Absorbing Fabric and Wave-Transparent Fabric

**DOI:** 10.3390/polym14030377

**Published:** 2022-01-19

**Authors:** Jianjun Yin, Wensuo Ma, Zuobin Gao, Xianqing Lei, Chenhui Jia

**Affiliations:** School of Mechatronic Engineering, Henan University of Science and Technology, 48 Xiyuan Road, Luoyang 471003, China; jianjun1010@126.com (J.Y.); gaozuobinly@126.com (Z.G.); lxq@haust.edu.cn (X.L.); 9903170@haust.edu.cn (C.J.)

**Keywords:** electromagnetic shielding fabrics, wave-absorbing fabrics, wave-transparent fabrics, mechanisms, research status, fabrics structure

## Abstract

As the basic materials with specific properties, fabrics have been widely applied in electromagnetic (EM) wave protection and control due to their characteristics of low density, excellent mechanical properties as well as designability. According to the different mechanisms and application scenarios on EM waves, fabrics can be divided into three types: EM shielding fabric, wave-absorbing fabric and wave-transparent fabric, which have been summarized and prospected from the aspects of mechanisms and research status, and it is believed that the current research on EM wave fabrics are imperfect in theory. Therefore, in order to meet the needs of different EM properties and application conditions, the structure of fabrics will be diversified, and more and more attentions should be paid to the research on structure of fabrics that meets EM properties, which will be conductive to guiding the development and optimization of fabrics. Furthermore, the application of fabrics in EM waves will change from 2D to 3D, from single structure to multiple structures, from large to small, as well as from heavy to light.

## 1. Introduction

As a carrier of information transmission, EM waves transmission have been exploded with the development of electronic equipment and communication technology [1]. As shown in Figure 1, according to the different frequencies and wavelength of EM waves, in the range of 10^3^–10^14^ Hz EM waves can be divided into radio waves, microwaves and infrared, and the generated radiation is non-ionizing radiation [2]. Frequencies between 10^15^–10^22^ Hz can be divided into ultraviolet rays, X-rays and gamma rays, and the generated radiation is ionizing radiation [3]. EM waves with different frequencies are applied in different scenarios and affect the development of human society to different degrees. On the one hand, the thermal and non-thermal effects of EM waves on biological tissues will lead to damage to the human body, such as fatigue, insomnia, tension, headaches and other symptoms will occur in the human body, and even increase the risk of leukemia and tumors under long-term exposure to EM radiation [4,5,6]. On the other hand, high frequencies of EM waves will interfere with the normal operation of electronic equipment and communication, even damage equipment and cause communication information leakage [7,8]. EM shielding materials, wave-absorbing materials and wave-transparent materials are the key to control EM waves transmission and pollution [9,10,11]. The development of EM materials that can work at wide frequencies, have low density, thin thickness and good protective performance are a hot spot of current research [12,13]. Among them, fabric is favored by researchers for its low density, excellent mechanical properties and structure diversified.

It is called fabrics that the fibers or fibers bundles with specific properties are arranged and combined in 2D or 3D space according to specific rules to form a collection of fiber bundles with a certain thickness, shape and geometric size by textile technology [14]. As base materials with specific properties, fabrics can be made into fabric reinforced composites, fabric preforms and other fabric-based composites [15,16]. According to the different fabric structures, it can be divided into woven fabrics, braid fabrics, knitted fabrics, etc., as shown in Figure 2 [17,18,19,20,21]. Figure 2a is a 3D angular interlocking woven fabric, the red and yellow in which represent yarns of different warp and weft directions. Figure 2b is a 3D braided fabric based on the symmetry of space group P4, different colors in which represent different paths of braided yarns. Figure 2c is a kind of knitted fabric, green and blue in which represent different types of coils that can be connected with each other to form a knitted fabric. Due to its low-density gravity and excellent mechanical properties, fabrics are widely used in aerospace, transportation, building construction, EM waves protection and other fields [22,23,24]. Fabrics also have excellent designability, by changing fibers characteristics, fabrics structures, fabrics layers, yarn density and direction Angle (or braiding Angle) to meet the ideal application conditions [25]. With the rapid development of fibers metallization, fibers modification and conductive polymer polymerization, fabrics have been widely used in the field of EM waves. A series of fabrics composites with EM function, flexibility and light weight have been formed [26,27,28]. According to the specific application scenarios, fabrics that meet the EM properties can be divided into EM shielding fabrics, wave-absorbing fabrics and wave-transparent fabrics.

The mechanisms of fabrics on EM waves can be divided into the macro level and the micro level. In macro level, Maxwell’s equations revealed the relationship between electric fields and magnetic fields, as well as the interaction law between fields and mediums by mathematical methods [29]. A changing electric field gives rise to a changing magnetic field, which in turn gives rise to a changing electric field, so that the magnetic field alternates with the electric field, excites each other, and travels away at a certain speed, producing electromagnetic waves [30,31]. Based on Maxwell’s equations, scholars have deduced many mathematical theories and methods for the interaction between EM waves and mediums, among which, transmission line theory has become a mainstream analysis method with easy understanding, convenient calculation and high precision [32,33]. Figure 3 shows the mechanisms of EM waves on material based on transmission line theory, including reflection loss on material surface, absorption loss of material and multiple reflection loss inside the material.

In micro level, skin effect, polarization and magnetic loss will occur when EM waves interact with materials. Skin effect will cause thermal effect caused by an increase of conductor resistance due to uneven current in the conductor [34]. Polarization will cause Debye relaxation and dielectric loss and the loss is related to imaginary part of the complex dielectric constant of materials [35]. The magnetic loss will produce resonance loss and eddy current loss and other losses, which is related to loss Angle tangent and imaginary part of complex permeability [36]. According to specific application scenarios, fabrics with EM properties can be divided into EM shielding fabrics, wave-absorbing fabrics and wave-transparent fabrics. The following describes the research status of these three fabrics, respectively.

## 2. EM Shielding Fabrics

As one of the effective means to restrain EM interference and realize EM protection, EM shielding means to limit the transmission of EM energy from one side of the material to the other side [37,38]. The mechanisms of EM shielding can be analyzed by transmission line method. Materials with high conductivity are usually used to restrain EM radiation, with the reflection effect of conductor on EM waves. Shielding effectiveness (*SE*) is usually used to represent the shielding ability and effect of materials on EM [39,40].

Although traditional metals and alloy materials have a good EM shielding effect, their development is limited by the disadvantages of heavy weight, high cost and poor corrosion resistance. Novel EM shielding materials with lightweight characteristics are becoming more and more popular. EM shielding fabrics have the advantages of low density, good flexibility and light weight, which are widely used in the manufacture of EM protection products such as protective clothing, shielding tents and shielding gun-suit [41]. EM shielding fabrics also have strong one-time molding ability, excellent designability, breathable fabrics properties, both soft and EM shielding properties, which can be made into different geometry to shield radiation source, but also can be processed into shielding suit and shielding cap to make the staff from EM radiation [42]. In addition, metal fibers fabrics also have other functions, such as antistatic, antibacterial and deodorant. EM shielding fabrics are the ideal shielding materials with outstanding properties. The research of EM shielding fabrics can be divided into theoretical calculation and experimental measurement.

### 2.1. Theoretical Calculation of EM Shielding Fabrics

The expression to measure shielding effect of materials are transmission coefficient *T* and Shielding Effectiveness (*SE*). The transmission coefficient *T* is the ratio of electric field intensity *E_t_* (or magnetic field intensity *H_t_*) at a place with a shield to electric field intensity *E_0_* (or magnetic field intensity *H_0_*) at the same place without a shield and the formula is as follows,
(1)$$T=\frac{E_{t}}{E_{0}}=\frac{H_{t}}{H_{0}}$$

*SE* refers to the shielding capability and effect of a shielding body against EM interference, which is often expressed logarithmically, as defined below [37],
(2)$$SE=20\mathrm{lg}\left(\frac{E_{0}}{E_{t}}\right)=20\mathrm{lg}\left(\frac{H_{0}}{H_{t}}\right)=10\mathrm{lg}\left(\frac{P_{0}}{P_{t}}\right)=20\mathrm{lg}\frac{1}{\left|T\right|}$$
where $\mathrm{lg}=\log_{10}$; $P_{0}$ is the power density without shielding; $P_{t}$ is the power density with shielding body at the same place. For the convenience of calculation, the most used formula is $SE=20\mathrm{lg}\left(\frac{E_{0}}{E_{t}}\right)$.

According to Figure 3, *SE* can be composed of reflection loss $SE_{R}$, absorption loss $SE_{A}$ and multiple reflection loss $SE_{M}$.
(3)$$SE=SE_{R}+SE_{A}+SE_{M}$$

The current theoretical calculation methods of EM shielding fabrics are to directly equivalent the conductive yarns with shielding performance to metal plates, and then the equivalent calculation is carried out according to the fabric structures corresponding to metal plate structures, such as no pore, pore structure, metal grid, layered parallel array and other structures, so as to calculate *SE* of fabrics. Based on transmission line theory, there are three different mechanisms for EM waves attenuation by the shielding body: reflection attenuation, absorption attenuation and multiple reflection attenuation. Firstly, metal plates are classified into no pore, pore structure, metal grid, layered parallel array (as shown in Figure 4), then the theoretical formulas or semi-empirical formulas of EM shielding are derived based on transmission line theory and equivalent circuit methods [43,44,45,46].

According to the literature [47,48], under the condition of far-field plane waves, transmission coefficient of no pore metal plates is as follows,
(4)$$(\begin{array}{l} T=\frac{4\eta_{e}\eta_{m}e^{-\gamma d}}{{\left(\eta_{e}+\eta_{m}\right)}^{2}}/[1-{\left(\frac{\eta_{m}-\eta_{e}}{\eta_{m}+\eta_{e}}\right)}^{2}\cdot e^{-2\gamma d}]\\ \eta_{m}=3.69\times 10^{-7}\sqrt{\frac{f\mu_{r}}{\sigma_{r}}}\\ \eta_{e}=377\Omega \\ \gamma =(1+j)\sqrt{\pi \mu f\sigma } \end{array}$$
where $\eta_{e}$ is the impedance of EM wave; $\eta_{m}$ is the impedance of the metal plate; γ is the propagation constant of EM wave in metal; *d* is metal plate thickness; $\mu_{r}$ and $\sigma_{r}$ are relative permeability and relative conductivity of metal plate; μ and σ are permeability and conductivity of metal plate. Combined with Equation (2), the *SE* of metal plate without poles can be obtained.

As for pore structure metal plates, the transmission coefficient of pores $T_{h}$ can be obtained according to the literature [49],
(5)$$T_{h}=4n{\left(\frac{q}{F}\right)}^{3/2}.\;(\mathrm{Circular}\;\mathrm{pore})$$
(6)$$T_{h}=4n{\left(\frac{kq^{’}}{F}\right)}^{3/2}.\;(\mathrm{Rectangular}\;\mathrm{pore})$$
where q is the area of a single circular pore; $q^{’}$ is the area of a single rectangular pore; *n* is the number of holes; *F* is the metal plate area; $k=\sqrt[{3}]{\frac{b}{a}\xi^{2}}$, *a* and *b* are short and long sides of rectangle pore, respectively; when the rectangle is square, ξ=1; when $\frac{b}{a}\gg 5$, $\xi =\frac{b}{2a\ln\frac{0.63b}{a}}$. The transmission coefficient of pore structure metal plate is $T_{h}=T+T_{h}$. Combined with Equation (2), *SE* of metal plate with pole structure can be obtained.

Henn et al. [50] first proposed that metallized fabrics were regarded as pores structure metal plates, and deduced the *SE* formulas of metallized fabrics by calculating the value of pores structure metal plates *SE*. Safarova et al. [51] used the above method to calculate metal fibers blended fabrics, analyzed the fabrics pore shape with image processing technology, approximated the irregular shape into a rectangle, and established the *SE* model about fabrics porosity, thickness and fibers volume.

The method of equivalent metal yarns to pores structure metal plates provides an idea for solving the *SE* of EM shielding fabrics. However, these models have certain limitations, requiring that the whole fabric has good electrical connectivity, resistance must equal to that of metal plates, fabrics must have a certain thickness, and the pores in fabrics need to be regular. In addition, simply approximating the shape of a single pore to a rectangle or a circle will cause a large error. When metal fibers content is too low, these models are not applicable, which is not conducive to the development of EM shielding fabrics.

*SE* formulas of metal mesh can be obtained from Literature [52].
(7)$$SE=A_{a}+R_{a}+B_{a}+K_{1}+K_{2}+K_{3}$$
where $A_{a}$ is absorption loss of pores; $R_{a}$ is reflection loss of pores; $B_{a}$ is multiple reflection loss; $K_{1}$ is the modification item related to the unit area and the number of pores $K_{2}$ is the modification item related to skin depth; $K_{3}$ is the modification item for coupling of adjacent pores. The calculation formula of each item was shown in Table 1, and the data was derived from [52].

It is difficult to accurately calculate the *SE* of metal grids. For the convenience of calculation, under approximate conditions, the *SE* of metal materials with good electrical conductivity mainly comes from reflection loss, and the absorption loss can be ignored. Engineering calculation of *SE* can be obtained that [53],
(8)$$SE=20\mathrm{lg}\frac{1}{s\sqrt{{\left[0.265\times 10^{-2}R_{f}\right]}^{2}+{\left[0.265\times 10^{-2}X_{f}+0.333\times 10^{-8}f(\ln\frac{s}{a}-1.5)\right]}^{2}}}$$
where *s* is the pitch of the metal grid; $R_{f}$ is AC resistance per unit length of metal grid; a is metal fibers radius; $X_{f}$ is the reactance per unit length of the metal grid.

Chen et al. [54] made polypropylene fibers woven with copper wire and stainless-steel wire conduct fabrics, respectively, proposed the metal grid structure, and calculated conduct fabrics SE by using the formulas of metal grid structure from the literature. In the frequencies range of 30 MHz–1.5 GHz, the measured values were quite different from theoretical values, which may be caused by poor contact or low conductivity of fabrics at yarn intersections. Cai et al. [55] used a metal grid structure model to calculate the *SE* of stainless-steel fibers blended fabrics. When the content of stainless-steel fibers was 5%, 10% and 15%, respectively, the calculated results were close to experimental results under low frequencies conditions. Rybicki et al. [56] established an equivalent circuit model of conductive grid yarns based on a periodic metal grid structure, believing that *SE* depends on grid size, thickness and resistivity of grid material. Compared with simulation experiments, this method had certain feasibility.

Although the structure of metal mesh is close to real 2D fabrics in shape, the method requires that the intersecting points of fabrics grid should be conductive, the pores should be regular, and the content of conductive fibers should not be too low. Moreover, the yarns containing metal fibers are a mixture of metal fibers and other fibers, which will affect its EM parameters and cause large errors. This method is not suitable for the large degree of buckling or 3D fabrics, which will limit the development of EM shielding fabrics structure to a certain extent.

Other optimization methods to calculate SE include Sabrio’s metal parallel array method, as shown in Figure 4 [57]. The metal grid was divided into two periodic arrays of parallel metal plates with different angles, and SE of each periodic array metal plate can be calculated. Liang et al. [58] derived a SE model of 2D metal fibers blended woven fabrics base on this method. According to the comparison between theoretical values and measured values, yarn diameter, electrical conductivity and weaving Angle all have a certain influence on SE. Whether the fabric is conductive at the yarn crossing point has no effect on this model, which has high applicability.

Yin et al. [59] established the SE model of plain weave fabrics by the way of the weighted average based on fabrics buckling surface equation and fabrics structure. This model explained the mathematical relationship between SE and the parameters of plain weave fabrics such as pitch, thickness and fiber volume content. The trend of this model was basically consistent with the experiment, which provided a theoretical reference for the effective design of EM shielding fabrics with a large degree of buckling.

The metal yarn was equivalent to the structure of no pores, pores, metal grid and so on, requiring yarn crossing point conductive, and fabrics need to have a certain thickness, which will limit fabrics design and development to a certain extent and there will be considerable limitations. The method equivalent to parallel metal array structure was more accurate and had no effect on whether the yarn crossing point was conductive or not, but this model was not suitable for 2D fabrics with a large degree of buckling and 3D fabrics. At present, the research on *SE* are limited to 2D fabrics, and there are few reports on 3D fabrics. 3D fabrics have greater development potential and stronger functions than 2D fabrics. The study of the influence of fabrics structure on *SE* will be the theoretical guiding significance to the development of 3D EM shielding fabrics.

### 2.2. The Experiment to Investigate EM Shielding Fabrics

The development of EM shielding fabrics can be generally processed by the method of surface metallization, metal coating or woven fabrics with conductive fibers, and then to investigate the influence of various parameters of EM shielding fabrics through experimental measurement.

The method of metal coating: Li et al. [60] conducted experimental tests on SE of silver-plated fiber fabrics, copper-nickel fiber fabrics and stainless-steel fiber blended fabrics. The results showed that, at the same frequency, the shielding effect of vertical polarization wave direction and horizontal or 45 degree polarization wave direction of silver-coated fiber fabric and the copper-nickel fabric was higher than that of stainless steel fiber blended fabrics, and the higher the folding degree, the greater the *SE*. Duan et al. [61] coated stainless steel electromagnetic shielding fabrics with carbon nanotubes, graphene, ferrite and nano nickel powder, respectively, and studied the effect of double-layer mixed coating on EM shielding fabrics. The test results showed that the best shielding effect was the double layer combination of graphene + ferrite and graphene + nickel, and the higher the coating thickness, the better the *SE*.

The method of surface metallization: Cheng et al. [62] experimentally studied the influence of different weft densities, warp densities, wire diameters and layering angles on EM shielding effect of copper-coated twill fabrics, and concluded that the number of conductive layers, warp and weft densities were positively correlated with *SE*. Liu et al. [63] prepared Ni/PPy (polymerization of pyrrole)/PET (polyethylene terephthalate) conductive fabrics with EM shielding effect by in-situ polymerization of pyrrole and electroless nickel plating, which had the abilities of flexible, lightweight and breathable. Conductive fabrics with higher fractal dimensions have higher thickness of conductive layer, higher conductivity and better EM shielding effect.

The method of braided fabrics with conductive fibers: Lopez et al. [64] explored the influence of metal fibers content on *SE*, studied the influence of EM waves frequencies and fabrics warp and weft density changes on *SE*, and showed that metal content index was positively correlated with *SE*. There was a negative correlation between electromagnetic frequencies and *SE* when metal content is constant. When fabrics metal index was the same, *SE* will not change regardless of yarn density. Liu et al. [65] conducted experiments on different types of EM shielding fabrics, and concluded that under the same parameters, *SE* of plain weave fabrics was better than that of twill weave fabrics, and *SE* of twill weave fabrics was better than that of satin weave fabrics, and indicated that fabrics porosity was the key factor to affect the SE. Liu then used the surface digital image analysis technology to analyze the surface of metal fabrics and establish the characteristic matrix, and found that the percentage content of metal fibers, porosity and arrangement direction all had a great influence on *SE* [66]. Yang et al. [67] tested various parameters of stainless-steel fibers blended fabrics on the influence of EM shielding. The results showed that stainless steel fibers content of fabrics was positively correlated with *SE*, fabrics compactness was negatively correlated with *S*, fabric with a small difference in warp and weft density showed better *SE*, and the *SE* of bidirectional blended yarns were better than that of unidirectional blended yarns.

A large number of experiments have qualitatively studied the factors affecting the EM shielding fabrics, and the general influence factors were the material properties, EM wave frequencies, polarization direction, metal yarns EM parameters, metal yarns arrangement spacing and arrangement, the coating thickness of metallizing fabrics, conductive yarns percentage content and porosity, etc. There are a few research on the influence of fabrics structures on EM shielding. There are many kinds of fabrics, and it cannot be ignored that the influence of different shielding fabrics structure on EM waves. With the diversity of fabrics structure, EM shielding fabrics will also develop in the direction of diversification. The research of 3D fabrics SE model or the influence of fabric structure on EM waves could be very important for the development and optimization of EM shielding fabrics.

## 3. Wave-Absorbing Fabrics

Wave-absorbing materials can effectively absorb EM radiation, reduce EM pollution, protect the ecological environment, protect all kinds of electronics and electrical equipment from EM interference, avoid equipment failure or aging, maintain the normal operation of equipment, and can provide effective protective measures for the human body. It is one of the important ways to control EM waves transmission and prevent EM waves pollution to prevent the human body from being harmed by EM radiation in a strong radiation environment [68,69]. With the rapid development of electronic information technology, the application of EM wave-absorbing materials is not limited to stealth military, but deep into communication anti-interference, electronic information confidentiality, environmental protection, human protection and many other fields [70]. The application of fabrics in the manufacture of wave-absorbing materials has the characteristics of good electromagnetic absorption capacity, strong designability, low manufacturing difficulty and low cost, and has high application value, such as the manufacture of aircraft fuselage skin, aircraft engine and radar stealth military tent [71].

### 3.1. The Mechanism of Wave-Absorbing Fabrics

In the design of wave-absorbing materials, in order to maximize the use of absorbing materials, the metal substrate is usually added to the bottom of the material to achieve strong reflection. When the transmitted waves incident on the surface of the metal substrate, it will be reflected back to wave-absorbing material for absorption loss, as shown in Figure 5.

Based on transmission line theory, it can be obtained according to the literature [72],
(9)$$\left\{ \begin{array}{l} R(dB)=20\log\left|\frac{Z_{in}-Z_{0}}{Z_{in}+Z_{0}}\right|\\ Z_{in}=Z_{0}\sqrt{\frac{\mu_{g}}{\varepsilon_{g}}}\tanh\left[j(\frac{2\pi fd}{c})\sqrt{\mu_{g}\varepsilon_{g}}\right]\\ Z_{0}=\sqrt{\frac{\mu_{0}}{\varepsilon_{0}}} \end{array}\right.$$
(10)$$\left\{ \begin{array}{l} \varepsilon =\varepsilon^{’}-j\varepsilon^{’’}\\ \mu =\mu^{’}-j\mu^{’’}\\ \tan\delta_{\varepsilon }=\varepsilon^{’’}/\varepsilon^{’}\\ \tan\delta_{\mu }=\mu^{’’}/\mu^{’} \end{array}\right.$$
where $Z_{in}$ represents input impedance; $Z_{0}$ represents free space impedance; *f* represents frequency; *c* represents the propagation speed of EM waves in a vacuum; *d* represents material thickness; $\mu_{g}$ and $\varepsilon_{g}$ represent complex permeability and permittivity of absorbing material, respectively; $\mu_{0}$ and $\varepsilon_{0}$ represent complex permeability and permittivity in free space. $\varepsilon^{’}$ and $\varepsilon^{’’}$ represent real and imaginary parts of complex permittivity; $\mu^{’}$ and $\mu^{’’}$ represent the real and imaginary parts of complex permeability; $\tan\delta_{\varepsilon }$ and $\tan\delta_{\mu }$ represent the tangent of dielectric loss Angle and magnetic loss Angle.

It can be seen that the material parameters affecting absorbing performance mainly include complex dielectric constant, complex permeability and loss Angle tangent. As for wave-absorbing fabrics, the absorbing effect is mainly achieved by absorption loss of fabrics to EM waves. The key factor to calculate absorption rate is to figure out the EM parameters of wave-absorbing materials. Since wave-absorbing fabrics with the inhomogeneous structure, most scholars’ method are to equalize a inhomogeneous structure to the homogeneous structure, and calculate the equivalent EM parameters by theoretical or semi-empirical formulas, such as Maxwell-Garnet equivalent formulas, Bruggeman efficient medium theory and strong fluctuation theory [73]. Then the absorptivity of the fabric can be obtained according to transmission line theory. Peng et al. [74] used strong fluctuation theory to solve the equivalent EM parameters of 2D plain weave fabric composites. Yin et al. [75] solved the equivalent EM parameters of 3D woven fabrics based on the theoretical strong fluctuation theory and established the wave-absorbing model of 3D woven fabrics. The relationships between the degree of buckling, the shape of yarns cross section, the thickness of fabrics, the fiber volume fraction and absorptivity of 3D woven fabrics were explained mathematically.

### 3.2. Researches of Wave-Absorbing Fabrics Preparation

For the research of wave-absorbing fabrics development, the fabrics can be divided into coated wave-absorbing fabrics and structural base type wave-absorbing fabrics. Coated wave-absorbing fabrics mean the inside or surface of 3D fabrics are coated with wave absorbents, such as carbon black, graphite and ferrite resistance or magnetic medium materials. By separating wave absorbent materials and fabric stable structure, 3D structure fabric as the stable support structure and wave absorbent play a role in absorbing EM waves. Xie et al. [76] embedded carbon black (CB) into 3D woven fabrics composite as the wave absorbent. The results showed that the absorbing performance of composite was significantly improved. The introduction of 3D woven fabrics can reduce the complex dielectric constant of composite, thus improving the impedance matching of composite, reducing the reflection of EM waves and improving absorbing performance. Zou et al. [77] coated carbon nanotubes (CNTs) on NaOH-pretreated cotton fabrics by the method of non-adhesive dip coating. The surface morphology and modification of carbon nanotube functionalized fabrics were studied by scanning electron microscopy (SEM) and infrared spectroscopy. The effects of impregnation coating quantity, carbon nanotube concentration and impregnation temperature on the electrical conductivity, EM shielding effect and absorbing efficiency of cotton fabric were studied. The measurement results show that the absorption rate of EM wave was 65.7% by adding multilayer laminated fabric. Simayee et al. [78] mixed micromagnetic carbonyl iron powder with nano carbon black as the wave absorbent and coated it on polyester fabrics by the way of filling-drying curing and aluminum sputtering coating. The experimental results showed that the aluminized polyester fabrics coated with carbonyl iron powder and nano carbon black have better absorbability than those without aluminized polyester fabrics. Liu et al. [79] chose polyester woven fabrics as the basic fabrics. Ferrite and silicon carbide are wave absorbent at the bottom and surface, respectively. By optimizing EM parameters, the ferrite/sic double-coated polyester fabrics with absorbing properties were prepared. The results showed that the fabric had the best absorption performance at the frequency of 10 GHz.

The type of structural base wave-absorbing fabrics refers to the yarns or fiber bundles with wave-absorbing properties, such as nickel-iron fibers or carbon fibers are directly woven or woven into 3D fabrics based on stable structure, and then the wave-absorbing properties and influence parameters of 3D fabrics are tested by experiments. Ayan et al. used a vector network analyzer to conduct experimental tests on cotton fabric, carbon fabric and cotton-carbon fabric composites board within 3–18 GHz. The mechanical value of cotton fabric composite board was lower, but the EM wave absorption value in a certain frequency range was higher than that of carbon fabric composite board. Cotton-carbon fabric composites have better absorbing performance than pure carbon fabric composites in 12~18 GHz frequencies [80]. Fan et al. [81,82] tested three kinds of 3D woven carbon fiber/epoxy composites with different structures, and the experimental test results showed that the composite has good EM absorption and shielding efficiency, and its excellent mechanical properties and absorption capacity can be widely used in radar absorption structures. Xue, L et al. [83] studied 3D isotropic braided carbon fibers/glass fibers (CF/GF) bismaleimide composites and tested their EM absorbing properties under the condition of thermal oxygen aging. The results showed that the composites have better EM absorption properties than those without aging. Since the surface of the aged composite was not smooth due to thermal oxygen aging, which will cause a large number of cracks and voids. These cracks and voids caused more EM waves to react with the material surface, thus improving the absorbing performance. Tak, J et al. [84] proposed a wearable metamaterial microwave absorber, embedding two square ring resonators into the conductive fabric with a thickness of 1mm for indoor radar clear applications. At a specific frequency, the absorption peak was greater than 90%, and it had a good deformation effect, which can be easily worn on the body. Alonso-gonzalez et al. [85] fabricated a kind of frequency-selective surface 3D woven fabric, in which conductive yarns were woven into a cruciform frequency-selective surface. Due to the symmetry of surface, wave absorption performance was largely independent of polarization and incident Angle, which can achieve large broadband wave absorption. Compared with the traditional frequency selection surface, this type of frequency selection surface was more flexible and convenient, and provided the possibility of large-scale production. Bi et al. [86] prepared carbonyl iron/reductive graphene oxide /non-woven fabrics composite by the method of in-situ synthesis, which has excellent microwave absorption performance in the 2.91–5.1 GHz band, and its qualified absorption bandwidth reaches 9.2 GHz. This kind of flexible lightweight fabric composite can be used as a potential material for wearable EM absorption coatings and devices.

At present, the research on absorbents are relatively mature, however the fabrics with absorbents are lack of theoretical bases, such as the amount of absorbent and the position of absorbent can only be measured according to experience or experiment. Moreover, taking fabrics as the basic structure will affect the EM parameters of whole materials, and the EM waves absorption frequency of one single type of absorbent is narrow, which is not conducive to the development of fabric with wide absorption frequencies, and absorbent has the risk of instability and easy to fall off. As for the research on structure-based absorbing fabrics, most scholars only focus on absorbing fiber materials, such as metal modification of fibers or measurement of electromagnetic parameters of fibers. There is still a lack of reports on how the fabric structure affects EM wave absorption. The development direction of wave-absorbing fabrics will be ‘thin, light, wide, strong’ ‘thin’ refers to the thickness is becoming smaller, ‘light’ refers to the mass is becoming smaller, ‘wide’ means that fabrics can work in the ultra-wide band of EM waves, ‘strong’ refers to the absorption performance, environmental resistance, temperature resistance and other aspects will be stronger. Different fabric structures have different effects on absorbing waves, and it is urgent to study the influence of fabric structures on wave absorption.

## 4. Wave-Transparent Fabrics

Wave-transparent materials play an important role in national defense and military, aerospace, national economy and other fields, such as the manufacturing of radar radome, wave-transparent wall, protective wall and so on [87,88]. The wave- transparent materials with high transmittance, low reflectivity and loss, good mechanical properties, good structural stability and fatigue resistance are the focus of recent research [89]. Fabrics have the advantages of simple structure, easy processing, strong designability, one-time molding, excellent mechanical properties and structural stability, and low manufacturing cost, which has great application potential in transmitting materials such as missiles, carrier rockets, aircraft, microwave towers, microwave relay station, communication antenna radome and antenna window radome, and transmitting wall manufacturing, etc. [90,91,92].

### 4.1. The Mechanism of Wave-Transparent Fabrics

As shown in Figure 6, when EM waves pass through the wave-transparent material, part of it will be reflected on the material surface, and a small amount of loss will be converted into heat energy after entering the material, then the rest of the EM waves will pass through the wave-transparent material. In the macro level, the interaction between EM waves and material can be divided into three parts, reflected, loss and transmitted. Equation (11) can be obtained.
(11)T+R+A=1
where *T* means transmission coefficient, *R* means reflection coefficient, and *A* means attenuation coefficient.

Based on empirical formula and transmission line theory [93], it can be obtained that,
(12)$$\left\{ \begin{array}{l} A=\frac{2\pi d\varepsilon \tan\delta }{\lambda {\left(\delta -\sin^{2}\theta \right)}^{1/2}}\\ R=\frac{{\left(\varepsilon -\sin^{2}\theta \right)}^{1/2}-\varepsilon \cos\theta }{{\left(\varepsilon -\sin^{2}\theta \right)}^{1/2}+\varepsilon \cos\theta }\\ {\left|T\right|}^{2}=\frac{{\left(1-R^{2}\right)}^{2}}{{\left(1-R^{2}\right)}^{2}+4R^{2}\sin^{2}\varphi } \end{array}\right.$$
where $\varphi =\frac{2\pi d}{\lambda }{\left(\varepsilon -\sin^{2}\theta \right)}^{1/2}$; *d* is the thickness of material; ε is the relative dielectric constant of material; tanδ is the dielectric loss Angle tangent of material; θ represents the Angle between the incident wave and material; λ is the incident wavelength.

According to Equation (12), the dielectric constant and loss Angle tangent of materials are the key factors affecting EM waves transmission. The materials with low dielectric constant, low loss Angle tangent and porosity should be selected as the wave-transparent materials [94,95,96].

### 4.2. Researches of Wave-Transparent Fabrics Preparation

The key way of preparing wave-transparent fabrics is to obtain the ideal wave-transparent EM parameters, the mainstream approaches are the way of fibers surface functionalization, and then woven those fibers into fabrics to become the bearing substrate. After that, it is reinforced with a modified resin base and finally forms the inorganic porous wave-transparent fabric composites with high temperature resistance [97,98,99,100]. Tang et al. [100] functionalized the surface of Kevlar (POSS-g-Kevlar@PDA, f-Kevlar) cloth by the method of dopamine/polyhedral oligomeric silsesquioxane (DA/POSS), and the corresponding f-Kevlar cloth/bisphenol A cyanate ester (BADCy) matrix wave-transparent laminated composites were prepared by the method of impregnation, lamination followed by mold pressing. The obtained transmission efficiency was 92.0%, with good mechanical properties and heat resistance. Gu et al. [101] conducted surface functionalization treatment on F-PBO (p-phenylene-2, 6-benzobisoxazole) fibers, and coated them with lysozyme on the PBO fibers surface. A kind of double cyanate ester F-PBO fibers wave-transparent laminated composite was prepared by the method of impregnation-winding-lamination. Its transmission efficiency was 93.6%, and it has good thermal stability and tensile strength. Zou et al. [102] prepared 2.5D SiNO fibers fabric and 2.5D SiNOf/BN wave transparent composite with good heat resistance and mechanical properties through borazine infiltration and pyrolysis process at 1400 °C. The fracture behavior of these composites was studied according to fibers residual strength, mechanical properties of in situ fibers and substrate and the fiber/substrate bonding strength. The results showed that the radial stress increases with the increase of processing temperature, which further improved the interface bonding strength. Liu et al. [103] modified PBO fibers with incorporation of a fluoride-containing linear interfacial compatibilizer and curing with cyanate ester. PBO fibers/ cyanate ester wave-transparent laminated composites are prepared. Compared with Gu J. et al. [101] ’s method, this material has better wave- transparent properties and mechanical properties. Su et al. [104] prepared 3D puffiness SiB/NO microfibers by the method of polyborosiloxane sol electrostatic spinning and NH3 pyrolysis at 1000 °C. The average dielectric constant and loss tangent of SiB/NO fibers were 4.44 and 0.0029, respectively. Its good morphology and properties make it a candidate material for wave-transparent ceramic composites.

At present, most of the related research is on the surface treatment or modification of fibers materials to obtain ideal EM parameters and mechanical properties. However, there is a lack of research on the influence of fabric structure. The arrangement mode of fibers and the structural change of fabrics will have a certain relationship to wave-transparent performance. The research on the influence of structure can push the wave-transparent fabrics composites transferred from 2D to 3D, which has a prospective guidance for wave-transparent materials and makes the wave-transparent more diversified.

## 5. Conclusions

Fabrics, which have been widely employed in EM waves field, can be divided into EM shielding fabric, wave-absorbing fabric and wave-transparent fabric based on different application scenarios. The development statuses of these three fabrics were analyzed and summarized. With regard to the EM shielding fabric, current theoretical methods aimed to equivalent the yarns with metal properties to metal plates, perforated metal plates or metal grid structure on the basis of the permutation and combination. In addition, those methods are suitable for single-layer or double-layer fabric composite materials, and it is required that the intersection points of fabric grid should be conductive, which has certain limitations. In detail, for 2D fabrics with a large degree of buckling or multi-layer fabrics, those methods will cause a large error. The fabrication of EM shielding fabrics can be processed by means of surface metallization, metal coating or woven fabrics with conductive fibers. However, the experiment without structure optimization will lead to both low efficiency and high production cost. In addition, the EM shielding mechanism and analysis methods of EM properties based on fabric structures are not only hardly reported, but also lack of optimum design method for shielding fabrics. Furthermore, the theoretical research on absorbing fabrics mainly aimed to calculate the EM parameters of fabrics base on various equivalent methods so as to obtain high loss Angle fiber materials. However, the EM parameters obtained by means of equivalent methods couldn’t meet the requirements of wide frequencies and ideal EM parameters. In terms of experiments, absorbing fabrics were mainly studied on fiber materials, in which the ideal complex dielectric constant and complex permeability could be achieved through surface treatment or modification of the fibers or fibers base. However, there are problems of high experimental costs and complicated procedures. In addition, there are few reports on theories based on the structures of fabrics, and different structures have different effects on absorbing performance. Therefore, current research should lay emphasis on the development of the design method of a fabric structure that meets the requirements of absorbing performance. Moreover, the theoretical methods of wave-transparent fabrics were mature. The key factors affecting wave-transparent properties were dielectric constant and tangent of loss Angle. At present, the research on the development of wave-transparent fabrics mainly involved how to reduce the dielectric constant and loss Angle tangent, and meet the mechanical properties and ablative properties simultaneously. There are few reports on the influence of fabric structure on wave-transparent properties. Therefore, the development and optimization of the wave-transparent fabric structure design can make the wave-transparent fabric structure more diversified.

In order to meet the needs of different EM properties, fabrics will be diversified, intelligent, as well as meet the direction of multi-working conditions, such as from 2D to 3D. 3D fabrics can not only enhance the mechanical properties and thermal stabilities of composite materials, but also reduce the vertical stratification, which can meet the diversified requirements of EM properties in structural design. The fabric structures with EM properties will be designed from large to small and from heavy to light. In addition, more and more attentions should be paid to the research on fabric structures that satisfy EM properties. In brief, by optimizing fabric structure design, the ideal EM parameters of EM fabrics that meet the requirements of total reflection, zero reflection or total transmission are expected to be obtained.

## Figures and Tables

**Figure 1 polymers-14-00377-f001:**
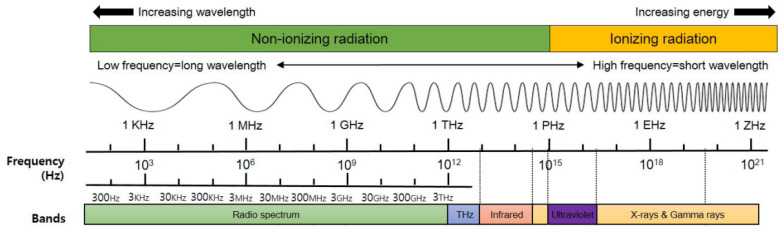
Schematic diagram of EM waves spectrum.

**Figure 2 polymers-14-00377-f002:**
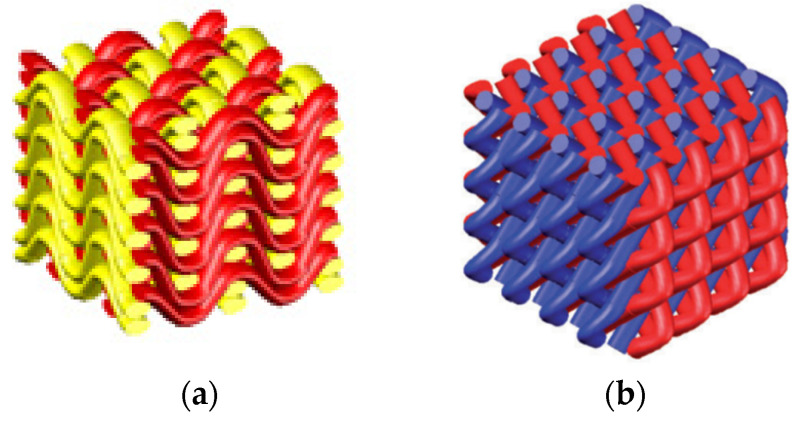
Schematic diagram of three kinds of fabrics. (**a**) is a 3D woven fabric; (**b**) is a 3D braided fabric [20]; (**c**) is a knitted fabric [21].

**Figure 3 polymers-14-00377-f003:**
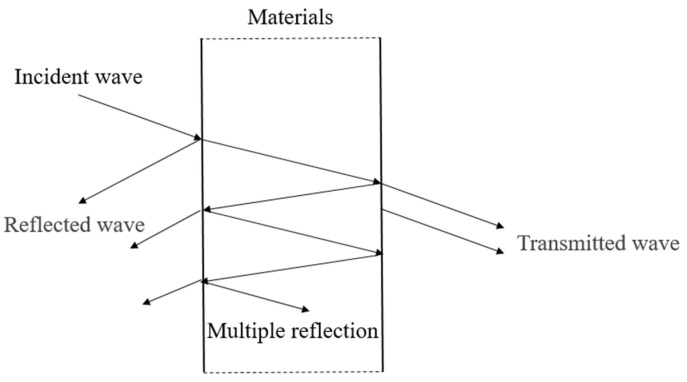
Schematic diagram of EM waves transmission on materials.

**Figure 4 polymers-14-00377-f004:**
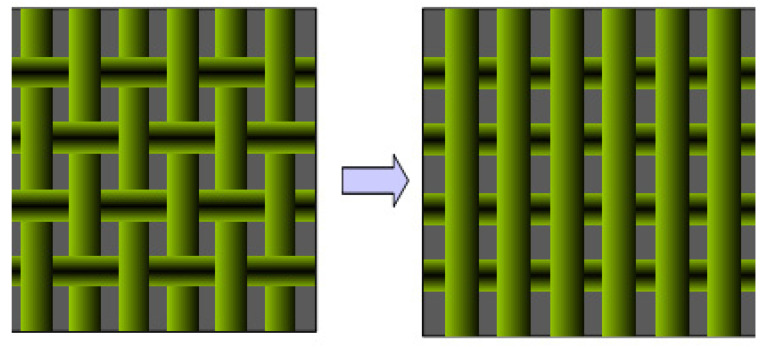
Schematic diagram of parallel array structures of metal.

**Figure 5 polymers-14-00377-f005:**
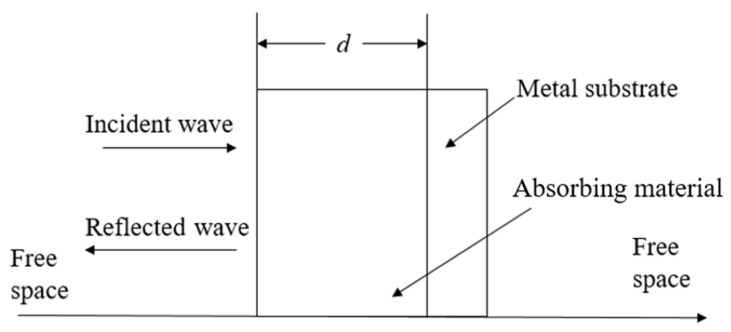
Schematic diagram of absorbing materials.

**Figure 6 polymers-14-00377-f006:**
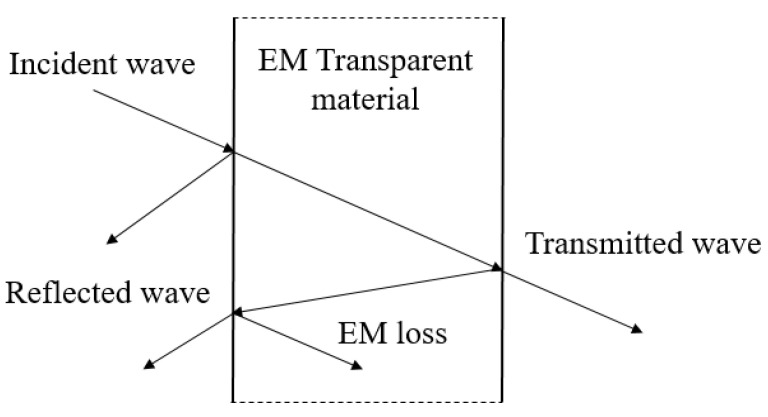
Schematic diagram of wave-transparent material mechanism.

**Table 1 polymers-14-00377-t001:** Formulas of metal plates with pore structure.

Symbols	The Calculation Formula	Instructions
$A_{a}$	$\begin{array}{l} 27.3\frac{d}{w},(\mathrm{rectangular});\\ 32\frac{d}{D},(\mathrm{circular}) \end{array}$	*d* is the depth of pores, cm; D is the diameter of a circular hole.
$R_{a}$	$20\mathrm{lg}\left|\frac{1+4K^{2}}{4K}\right|$	Rectangular pores: $K=j6.69\times 10^{-5}fw$ Circular pores: $K=j5.7\times 10^{-5}fw$
$B_{a}$	$20\mathrm{lg}\left|1-{\left(\frac{K-1}{K+1}\right)}^{2}10^{-0.1A_{a}}\right|$	*f*, MHz
$K_{1}$	−10lg(a⋅n),r≫w	*r* is the distance between shield and field source; *a* is the area of a single pore, cm^2^; *n* is the number of pores per square centimeter
$K_{2}$	$-20\mathrm{lg}(1+35p^{-2.3})$	$P=\frac{Width\;of\;conductor\;between\;holes}{Skin\;depth}$
$K_{3}$	$20\mathrm{lg}\left[\coth(\frac{A_{a}}{8.686})\right]$	———————————————

## Data Availability

All the data can be available in Reference.

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
