# Peer review of "A Review of Electromagnetic Shielding Fabric, Wave-Absorbing Fabric and Wave-Transparent Fabric"

_polymers, 2022, doi:10.3390/polym14030377_

Round 1

Reviewer 1 Report

A review of fabrics for electromagnetic waves 
Jianjun Yin, Wensuo Ma1, Zuobin Gao, Chenhui Jia and Xianqing Lei

This manuscript presents a review of fabrics intended to shield electromagnetic waves from the human body. This is an interesting
and hot topic, which surely deserves to be properly reviewed. Thus, the topic is well suited for Polymers. 
The present manuscript comprises 8 figures, 1 table, and 103 references are given.
The figures included here are mostly well done. However, most of the figures seem to stem from teaching books or research papers,
and I cannot find any copyright issue. This must be properly arranged prior to publication. Furthermore, all figure captions in this
manuscript are unsuitable. The figure captions should detail what one can see in the figure, and provide all information required
to understand the content of the figure. This really helps the readers a lot!
The most important point is the problem with the English language. The English requires substantial improvement. Please have a 
native speaker checking the manuscript. This checking will also reveal many other problems arising in this manuscript: This already
starts in the title which lacks the most important message -- for what use? Thus, the title of the manuscript should read like:
"A review of fabrics to shield electromagnetic waves".

Overall, the authors discuss the issues of EM wave transmission and shielding quite well, providing several respective formulae
describing the requirements and experiments carried out in the literature. This is well arranged, but eventually, the authors may
include graphs of important results in this review article as well.

There is a large amount of technical problems in this manuscript which require attention prior to publication:
(1) Please check the spaces in the entire manuscript. There should always be spaces between quantities and units, text
    and brackets and text and citations. In the present form, the manuscript is extremely difficult to be read.
(2) References are NOT numbered consecutively.
(3) Fig.2: Explain the colors.
(4) Fig.3 has nothing to do with transmission. This is a scheme showing the basic layout of an EM wave in general. Then, there should
   be a much better explanation to the figure, e.g., the greek letter nu is not used anywhere is this manuscript.
(5) Eq.2 uses the abbreviation lg, later on log is used as well. Please be consistent and mention at the first appearance which
   logarithm is used!
(6) Please take care for proper formatting. Sommetimes, words are capitalized without any sense.
(7) Chemical formulae and physical units MUST be properly formatted (e.g., MHz or NaOH)
   and abbreviations of chemical substances should be explained at the first occurence in the text.
(8) Text in a formula should be in roman.
(9) The lettering of Fig.5 is unreadably too small and blurred.
(10) When mentioning a first author's name in the text, remove the initials.
(11) Reference list: What are the [J] and [M] in the paper titles for?
(12) Reference list: There are no page limits, so no need to use "et al." anywhere.
(13) Reference list: Chemical formulae in the titles must also be formatted properly.
(14) Reference list: Please provide proper abbreviations of the journal names.

Overall, this manuscript may be interesting, but the hugh amount of technical deficits force a rejection -- in the present form,
this manuscript is not suitable for publication at all.   

Author Response

Dear reviewers and editors,

Thank you very much for your professional and patient reviews. Your opinions are of great help to our works. The comments have been carefully taken into account and a new revised submission have been uploaded.

  • References and titles were mainly modified.
  • The content of section 5 has been improved.
  • Many grammatical errors have been corrected and English writing has been improved.
  • Modified the figures in manuscript, and made the supplementary explanation.

We highlighted all the altered passages in red. The responses are as follows:

To reviewer 1:

To question 1: Please check the spaces in the entire manuscript. There should always be spaces between quantities and units, text and brackets and text and citations. In the present form, the manuscript is extremely difficult to be read.

Response: Thank you for your patient instruction. All spaces in article have been corrected.

To question 2: References are NOT numbered consecutively.

Response: References have been reordered.

To question 3: Fig. 2: Explain the colors.

Response: An explanation has been added in article. Fig. 2(a) is a 3D angular interlocking woven fabric, the red and yellow in which represent yarns of different warp and weft directions. Fig. 2(b) is a 3D braided fabric based on the symmetry of space group P4, different colors in which represent different paths of braided yarns. Fig. 2(c) is a kind of knitted fabric, green and blue in which represent different types of coils that can be connected with each other to form a knitted fabric.

To question 4: Fig. 3 has nothing to do with transmission. This is a scheme showing the basic layout of an EM wave in general. Then, there should be a much better explanation to the figure, e.g., the greek letter ‘nu’ is not used anywhere is this manuscript.

Response: Thank you for your patient instruction, Fig. 3 is trivial and according to reviewer 2 advice, Fig. 3 has been removed.

To question 5: Eq. 2 uses the abbreviation ‘lg’, later on ‘log’ is used as well. Please be consistent and mention at the first appearance which logarithm is used!

Response: An explanation has been added at the bottom of Eq. 2 about which logarithm is used. ‘lg’ is not the same as log:

To question 6: Please take care for proper formatting. Sometimes, words are capitalized without any sense.

Response: Thank you for your patient advice. Some explanation has been added before the capital letters.

To question 7: Chemical formulae and physical units must be properly formatted (e.g., MHz or NaOH) and abbreviations of chemical substances should be explained at the first occurence in the text.

Response: Thank you for your patient advice. Some explanation has been added before the abbreviations of chemical substances.

To question 8: Text in a formula should be in roman.

Response: It has been modified in type of roman.

To question 9: The lettering of Fig. 5 is unreadably too small and blurred.

Response: It has been replaced with a clearer figure.

To question 10: When mentioning a first author's name in the text, remove the initials.

Response: Thank you for your patient advice. It has been modified.

To questions 11-14:

Reference list: What are the [J] and [M] in the paper titles for?

Reference list: There are no page limits, so no need to use "et al." anywhere.

Reference list: Chemical formulae in the titles must also be formatted properly.
Reference list: Please provide proper abbreviations of the journal names.

Response:

Improper format was used earlier and the format of all references has been revised.

Reviewer 2 Report

The paper can be accepted after the following corrections:

  1. Abstract: text "small specific gravity" should be explained or removed.
  2. line 23: "technology0.". Please correct.
  3. line 35: reference not found.
  4. The selection of figures in figure 1 is not clear. In my opinion, this figure is not suitable for the scientific review. Please re-draw. Preferably, please explain the applications in form of text instead of icons.
  5. Figure 3 is trivial and should be removed.
  6. Figure 5 is unclear. Please re-draw to clearly present the idea. Please also develop the figure's caption
  7. Section 4. "Conclusion and prospect" should be developed. Please clearly state the most important outcome of the review. Please clearly specify the most significant gaps in the state-of-the-art, as well as the most important achievements of presented analyses.

Author Response

Dear reviewers and editors,

Thank you very much for your professional and patient reviews. Your opinions are of great help to our works. The comments have been carefully taken into account and a new revised submission have been uploaded.

  • References and titles were mainly modified.
  • The content of section 5 has been improved.
  • Many grammatical errors have been corrected and English writing has been improved.
  • Modified the figures in manuscript, and made the supplementary explanation.

We highlighted all the altered passages in red. The responses are as follows:

To question 1: Abstract: text "small specific gravity" should be explained or removed.

Response: It has been modified to "low density".

To question 2: line 23: "technology0.". Please correct.

Response: It has been modified and we have checked all the space.

To question 3: line 35: reference not found.

Response: Reference has been added.

To question 4: The selection of figures in figure 1 is not clear. In my opinion, this figure is not suitable for the scientific review. Please re-draw. Preferably, please explain the applications in form of text instead of icons.

Response: Fig. 1 has been reworked to make it more concise by removing redundant images.

To question 5: Fig. 3 is trivial and should be removed.

Response: Fig. 3 has been removed.

To question 6: Fig. 5 is unclear. Please re-draw to clearly present the idea. Please also develop the figure's caption.

Response: It has been replaced with a clearer figure.

To question 7: Section 4. "Conclusion and prospect" should be developed. Please clearly state the most important outcome of the review. Please clearly specify the most significant gaps in the state-of-the-art, as well as the most important achievements of presented analyses

Response: Thank you again for your professional and patient reviews. Section 4 has been improved and more detailed summaries have been added for different types of fabrics. The content is added to analyze the shortcomings of current researches and the direction of improvement.

Reviewer 3 Report

This manuscript is a mini-review on fabrics which are widely used for electromagnetic wave protection and control. The topic of the work is quite highlighted and actual. The introduction section clearly describes the state of the art in the specific field of research and it is appropriately supported by the relevant references including a very recent ones as well as the reviews in the field. The main part of the manuscript adequately represent the advances related to the topic. Conclusion section well summarizes the mini-review. The technical quality of the manuscript is high.

(A) The topic of this manuscript does not perfectly fits the scope of the journal; this reduce the overall impact of this work. Authors might consider to submit this manuscript to another more specialized journal.

(B) Not all the references in the Introduction section appears correctly; Refs [5-8] are missing instead there is as text <Error! Reference source not found.> Please correct.

[C] In the list references an unclear symbol <[J]> stands for the selected references. Please consider to delete it or explain.

(D) The style of the manuscript language should be carefully checked by the native speaker; this increase the overall clarity of the manuscript.

Summary: This mini-review can be considered for publication at Polymers after a minor revision as given.

Author Response

Dear reviewers and editors,

Thank you very much for your professional and patient reviews. Your opinions are of great help to our works. The comments have been carefully taken into account and a new revised submission have been uploaded.

  • References and titles were mainly modified.
  • The content of section 5 has been improved.
  • Many grammatical errors have been corrected and English writing has been improved.
  • Modified the figures in manuscript, and made the supplementary explanation.

We highlighted all the altered passages in red. The responses are as follows:

To reviewer 3:

To question 1: The topic of this manuscript does not perfectly fits the scope of the journal; this reduce the overall impact of this work. Authors might consider to submit this manuscript to another more specialized journal.

Response: This manuscript mainly introduces the research status of electromagnetic fabrics, which can be composed of polymer fibers and involves the field of polymer science. In my opinions, this manuscript is suitable for “Polymers”.

To question 2: Not all the references in the Introduction section appears correctly; Refs [5-8] are missing instead there is as text <Error! Reference source not found.> Please correct.

Response: Thank you for your patient reviews. It has been corrected.

To question 3: In the list references an unclear symbol <[J]> stands for the selected references. Please consider to delete it or explain.

Response: Improper format was used earlier and the format of all references has been revised.

To question 4: The style of the manuscript language should be carefully checked by the native speaker; this increase the overall clarity of the manuscript.

Response: Many grammatical errors has been corrected and English writing has been improved.

Round 2

Reviewer 1 Report

referee report 
polymers-1524745-peer-review-v2
A review of fabrics for electromagnetic waves 
Jianjun Yin, Wensuo Ma1, Zuobin Gao, Chenhui Jia and Xianqing Lei

This manuscript presents a review of fabrics intended to shield electromagnetic waves from the human body. This is an interesting
and hot topic, which surely deserves to be properly reviewed. Thus, the topic is well suited for Polymers. 

The authors have now presented a revised version. Many of the previous comments are properly addressed, so the reference
list is completely rewritten and also several parts of the main body of the manuscript. This is well done, but shows
partly again the problems concerning the English -- some corrections were made to text which was ok, but the corrections
are now wrong. Thus, the English checkup must be done by a professional, native English speaker. The title of the
manuscript should better be now: "A review of wave-absorbing and wave-transparent fabrics to shield electromagnetic 
waves".

Then, some more points are left:
-- The meaning of the logarithm "lg" must be explained in the manuscript.
-- I cannot see a solution to the problem that most of the figures seem to stem from teaching books or research papers,
  and I cannot find any copyright issue. This must be properly arranged prior to publication. 
-- I am still wondering about this point from the previous report: The authors may include graphs of important results
  of other groups in this review article as well -- why is this not done? Any reason?

So, to sum up: The corrections and changes made by the authors go in the right direction. If the minor points mentioned
will be solved, the manuscript will get suitable for publication, provided that the improvement of the English is done
professionally.

Author Response

Dear reviewers and editors,

Thank you very much for your professional and patient reviews. Your opinions are of great help to our works. The comments have been carefully taken into account and a new revised submission has been uploaded.

  • The title has been revised.
  • References have been added to some figures and equation, besides three new references have been added to ‘References’.
  • English have been improved, especially in ‘Abstract’ and ‘Conclusion and prospect’.

To question 1: The title of the manuscript should better be now: "A review of wave-absorbing and wave-transparent fabrics to shield electromagnetic waves".

Response: The title of this manuscript has been revised “A review of electromagnetic shielding fabric, wave-absorbing fabric and wave-transparent fabric”.

To question 2: The meaning of the logarithm "lg" must be explained in the manuscript.

Response:

An explanation has been added to the original text: “where lg=log10 ”.

To question 3: I cannot see a solution to the problem that most of the figures seem to stem from teaching books or research papers, and I cannot find any copyright issue. This must be properly arranged prior to publication. 
-- I am still wondering about this point from the previous report: The authors may include graphs of important results of other groups in this review article as well -- why is this not done? Any reason?

Response:

I am very sorry for the confusion caused by my negligence. References have been added to Figure 3, Figure 4, Figure 5, Figure 6, Figure 7 and Equation (8), besides three new references have been added to ‘References’.

To question 3: The English checkup must be done by a professional, native English speaker.

Response:

This manuscript has been revised by a professional English teacher, especially in ‘Abstract’ and ‘Conclusion and prospect’. All the modifications have been marked.

Reviewer 2 Report

The paper was corrected and can be accepted in the present form.

Author Response

Thank you very much for your professional and patient reviews. Your opinions are of great help to our works. With best wishes for a happy New Year! 

Round 3

Reviewer 1 Report

The revision is well performed. Concerning the English, the editing process will hopefully get out the last problems.

Thus, the manuscript may now be published in Polymers, provided that there are no copyright issues concerning the images.